# Carbonizing technology enables *Sanguisorbae Radix* to inhibit yeast-to-hypha differentiation and biofilm formation in *Candida albicans*

Xuxi Cheng[1,2☯], Jinyun Song[1☯], Qinglian Hu[2☯], Hongdan Wu[2], Bohui Song[2], Ruixiao Ma[2], Jinghan Gao[1], Yiwei Wang[2], Huangjin Tong[3]*, Wei Gu[2]*, Hongyu Zhao[1]*

**1** Department of Clinical Research Center, The Second Hospital of Nanjing, Affiliated to Nanjing University of Chinese Medicine, Nanjing, China, **2** School of Pharmacy, Nanjing University of Chinese Medicine, Nanjing, China, **3** Affiliated Hospital of Integrated Traditional Chinese and Western Medicine, Nanjing University of Chinese Medicine, Nanjing, China

☯ These authors contributed equally to this work
* njdie001@njucm.edu.cn (HZ); guwei@njucm.edu.cn (WG); tonghj@jsatcm.com (HT)

## Abstract

Sanguisorbae Radix (SR) has been employed as an herbal medicine over centuries. Charred SR (CSR), acquired via carbonization after the charred stir-frying of SR, demonstrates superior antimicrobial activity compared to SR. The aim of the study was to identify how carbonizing technology enhanced the ability of SR to inhibit the transformation from yeast to hypha and biofilm formation in C. albicans. In this paper, a vulvovaginal candidiasis (VVC) mouse model was used to evaluate the therapeutic effects. After CSR treatment, VVC mouse models nearly eliminated hyphal C. albicans adhering to the vaginal mucosa. The inhibitory activities of CSR on C. albicans biofilm formation and hyphal growth were assessed through quantitative biofilm analysis, morphological observations, and gene expression studies in vitro. Since the hyphal form signifies the initiation of biofilm development, this study confirmed CSR's remarkable inhibitory effect on C. albicans biofilm formation and hyphal growth. These effects were significantly weaker with SR. Additionally, the impact of carbonization on the composition of active compounds was analyzed. Carbonization significantly increased the content of ellagic acid (EA) and pyrogallic acid (PYG) by 7.44-fold and 28.09-fold, respectively. Both EA and PYG inhibited C. albicans biofilms and hyphal growth, with EA showing a more pronounced inhibitory effect. Finally, we concluded that carbonization technology enables SR to inhibit the yeast-to-hypha transition and biofilm formation in C. albicans by increase the levels of EA and PYG. EA was identified as the primary bioactive compound responsible for CSR's anti-biofilm effects.

**Data availability statement:** All relevant data are within the paper and its Supporting Information files.

**Funding:** This work was financially supported by Nanjing Health Science and Technology Development Fund Medical Key Technology Development Project (Grant No. ZKX22039), the First Phase Reserve Talent Project of The Second Hospital of Nanjing (Grant No. HBRCYL09), Medical Research Project of Jiangsu Province Health Commission in 2023 (H2023084), Advanced Training Program for Leading Personnel in Traditional Chinese Medicine in Jiangsu Province(Jiangsu Traditional Chinese Medicine Science and Education [2022] no.17), Zeng Bailin Esteemed Veteran Pharmacist Heritage Workshop of Jiangsu Province (Jiangsu Traditional Chinese Medicine Research and Education [2024] No. 4), Jiangsu Pharmaceutical Association Jin Peiying Fund Project (Grant No. J2021002), the Natural Science Foundation of Jiangsu Province (BK20231308), National Key Research and Development Program of China 'Research on intelligent recognition and production control technology for stir frying traditional Chinese medicine slices' (Grant No. 2023YFC3504200) and National Famous Traditional Chinese Medicine Expert Inheritance Studio Construction Project (Grant No. State Administration of Traditional Chinese Medicine [2022]75). There was no additional external funding received for this study beyond these.

**Competing interests:** The authors have declared that no competing interests exist.

## Introduction

In 2022, the World Health Organization (WHO) first included fungi in its priority pathogens list, categorizing *C. albicans* as a severe-grade pathogen among 19 fungal species identified as posing the most significant menace to public health. Vulvovaginal candidiasis (VVC), a commonly prevalent infectious disease affecting the female reproductive tract [1], is primarily caused by *C. albicans* and affects an estimated 100 million women globally each year [2]. However, *C. albicans* is typically a commensal fungus, with infections are considered opportunistic in the majority of cases [3]. The yeast form of *C. albicans* is initially non-pathogenic, while its transition from yeast to mycelial form is a critical event in pathogenesis. This morphological shift enables the fungus to express virulence factors, including adhesion and invasion-associated factors, which facilitate host tissue colonization and infection [4–8].

The accumulation of polymorphic structures, comprising yeast, pseudohyphae, and hyphal cells, results in the development of a biofilm. This complex, a polymorphic structure that can exceed a depth of 200 micrometers, significantly enhances resistance to antimicrobial drugs and evasion of immune surveillance [9]. One of the most clinically significant phenotypic changes observed in cells forming biofilms is their enhanced resistance to antifungal therapy [10]. Compared to planktonic cells, biofilm cells can exhibit up to a 1000-fold increase in resistance [11]. Consequently, hyphal formation is considered one of the most critical virulence factors of *C. albicans*. The transition from yeast to hyphal form is considered a critical step in the pathogenesis of *C. albicans* [12–14]. Previous studies have primarily focused on strategies to inhibit hyphal formation in fungal pathogens [15–17]. Additionally, in order to prevent the formation of biofilms, the transformation of yeast cells to hyphae should be prevented either [18].

*Sanguisorbae Radix* (SR), the root of *Sanguisorba officinalis* L., has been established as a well- recognized herbal medicine in China, Korea, and Japan for its medicinal applications. It is frequently mentioned in classical texts such as the *Shennong Herbal Scripture*. In these ancient Chinese documents, SR is noted for its use in treating leukorrheal disorders (S1 Table), while VVC is classified as a leukorrheal disease. This traditional application may be attributed to SR's potent antimicrobial properties. Extensive research has demonstrated that SR exhibits significant and broad-spectrum antimicrobial activity against more than ten pathogenic microorganisms, including *Staphylococcus aureus*, methicillin-resistant *Staphylococcus aureus*, *Pseudomonas aeruginosa*, *C. albicans* and *Trichophyton rubrum* [19,20]. Among numerous antimicrobial traditional Chinese medicines (TCMs), SR is recognized for its superior antimicrobial efficacy, which is comparable to, or even surpasses, that of *Sophora flavescens*, a well-known antimicrobial TCM [21].

Traditional processing methods of SR include roasting, frying, simmering, and treatments involving vinegar, wine, or charcoal. Among these, raw SR and its charcoal-processed form continue to be widely used today. Charred *Sanguisorbae Radix* (CSR) is produced through carbonization following the charred stir-frying of SR. Carbonization can break down the structure of drug compounds by elevating the temperature, converting some compounds into other substances with fewer side effects to achieve

the purpose of improving therapeutic effects [22]. Previous studies have shown that CSR exhibits superior antimicrobial activity compared to raw SR [23–25]. However, the mechanisms underlying CSR's enhanced efficacy await further clarification. This study aims to explore the impact of carbonization technology on the antifungal activity of SR, particularly its regulation on the yeast-to-hypha transformation and biofilm formation in *C. albicans*. Additionally, the study sought to identify the key bioactive components responsible for the improved pharmacological effects. These findings provide a scientific basis for developing a novel antifungal agent with potential reduced resistance for the treatment of VVC.

## Materials and methods

### Chemicals and reagents

Chromatographic-grade methanol and acetonitrile were sourced from TEDIA (USA), while chromatographic-grade formic acid was acquired from Aladdin (Shanghai, China). HPLC reference standards, including gallic acid, ellagic acid, methyl gallate, and pyrogallic acid, were supplied by Yuanye Bio-technology Co., Ltd. (Shanghai, China). Catechin was obtained from Herbpurify Bio-technology Co., Ltd. (Chengdu, China), and protocatechuic acid (purity ≥ 98%) was provided by Abphyto Bio-technology Co., Ltd. (Shanghai, China). Sabouraud dextrose agar (SDA) medium was purchased from Biowell Bio-technology Co., Ltd. (Shanghai, China), whereas Sabouraud glucose liquid medium (SDB) was procured from Solabao Biotechnology Co., Ltd. (Beijing, China). Fluconazole and clotrimazole were sourced from Macklin (Shanghai, China). Electron microscopy fixative came from Proteinbio (Nanjing, China). Hematoxylin and eosin (HE) staining kits and periodic acid-Schiff (PAS) staining kits were also supplied by Solabao Biotechnology Co., Ltd. (Beijing, China).

### Processing of herbs and preparation of extracts

SR decoction pieces were acquired from Haichang Traditional Chinese Medicine Group Co., Ltd. and authenticated by Professor Lu Tulin from the School of Pharmacy, Nanjing University of Traditional Chinese Medicine. The materials complied with the standards outlined in Part 1 of the 2020 edition of the *Chinese Pharmacopoeia.* According to previous researches [26,27], the SR decoction pieces were processed using a preheated automatic frying machine under the following conditions: rotation speed of 26 rpm, temperature of 250 °C, and duration of 16 minutes. CSR was subsequently ground into powder using a grinder, followed by sieving through a No.2 pharmacopoeial sieve to obtain fine powder. To prepare the CSR extract, the powder of CSR was placed in a stoppered conical flask, and 10 volumes of 75% ethanol was added. The mixture was soaked in deionized water at 50 °C for 12 hours. The solvent was then removed using a rotary evaporator at 50 °C, and the resultant extract was dried in an oven.

### *C. albicans* strain and growth conditions

*C. albicans* (CMCC(F)98001, Lot. D1134B-220913) was supplied by Luwei Technology Co., Ltd. All *C. albicans* strains were maintained in a cryoprotectant solution composed of 75% SDB and 25% glycerol (V/V) and stored at −80 °C. The strains were first cultivated on SDA plates and incubated at 37 °C in a thermostatic chamber for 24 hours. For liquid culture, the strains were transferred into SDB and incubated within a constant temperature shaking incubator set at 150–200 rpm. Following fungal activation, 1 mL activated suspension was diluted with sterile physiological saline to achieve the turbidity standard of a 0.5 McFarland, and subsequently subjected to two-fold serial dilutions with sterile water to make the final concentration reach $10^6$ CFU/mL [28].

### Construction of a mouse model of VVC

All procedures in this animal experiment were approved by the Experimental Animal Ethics Committee of Nanjing University of Traditional Chinese Medicine (Approval Number: 202305A035). SPF-grade female mice were sourced from the Qinglongshan Animal Breeding Farm (Nanjing, China). The mice were acclimated for one week under controlled

conditions: ad libitum access to food and water, 60% humidity, a temperature of 26 °C, and a 12-hour light-dark cycle. Three days prior to vaginal colonization with *C. albicans*, the mice were intraperitoneally injected with 1 mg/mL estradiol suspension once daily. Subsequently, 10 μL *C. albicans* suspension, prepared in physiological saline to a final concentration of approximately $1.0 \times 10^7$ CFU/mL, was inoculated into the vaginal cavity of estrogen-treated mice. The following day, vaginal lavage fluid of the mice was collected and streaked onto SDA medium, which was incubated at 37 °C for 24 hours in an inverted position [29]. The proliferation of *C. albicans* on the culture medium was used as an indicator to confirm successful model establishment.

## Grouping and administration

Upon successful establishment of the model, mice in the SR and CSR extract groups were intravaginally administered 10 μL of a diluted physiological saline solution containing 500 μg/mL of the extracts on days 1, 3, and 6 post-inoculation. The model group and the positive control group received parallel treatments with equal volumes of physiological saline or a 200 μg/mL clotrimazole diluted in physiological saline, respectively. Vaginal lavage samples were collected on days 1, 3, 5, and 7 to evaluate *C. albicans* colony counts. On day 7, the mice were sacrificed via cervical dislocation, and vaginal tissues were collected, fixed in 4% paraformaldehyde, and prepared for pathological sectioning and histological examination.

## Preparation of paraffin sections of mouse vaginal tissue

The vaginal tissue fixed in 4% paraformaldehyde solution was carefully trimmed and placed in a tissue embedding cassette. The samples were progressively dehydrated using ethanol solutions with escalating concentrations (70%, 80%, 90%, and 100%), with each step lasting 20 minutes. Following dehydration, the tissue was cleared in xylene and subsequently immersed in paraffin at 65 °C for hardening. The hardened tissue was embedded in paraffin, frozen and demolded at −20 °C. Sections were then prepared using a semi-automatic microtome, with a thickness of 4 μm [30].

## Hematoxylin and Eosin (HE) staining and Periodic Acid-Schiff (PAS) staining

HE staining, a widely used histological technique, allows for effective visualization of the basic morphology and pathological changes of the vagina. Sections were dewaxed and rehydrated. The slides were stained with hematoxylin for 3−5 minutes, then rinsed with tap water to remove excess stain. Differentiation was carried out briefly using 1% hydrochloric acid, followed by rinsing and neutralization in 0.6%−0.7% ammonia water to restore a blue color. After another rinse with running water, eosin was used for counterstaining. The slides were then dehydrated, mounted, and examined under a microscope to observe the vaginal tissue morphology.

PAS staining, primarily used to detect glycogen and *C. albicans* in the vaginal epithelium, was also performed. Sections were dewaxed and rehydrated, followed by staining in periodic acid solution for 15 minutes and rinsing twice with distilled water. The sections were then treated with Schiff reagent, rinsed with running water, and counterstained with hematoxylin to visualize cell nuclei. After dehydration and mounting, *C. albicans* infection was examined microscopically.

## Pathological injury score

The pathological damage of the mucosal epithelium was evaluated based on three criteria: varying degrees of keratinization, ulceration, and erosion, with each criterion scored on a scale of 1–9 points. The pathological damage of the submucosal tissue was assessed using indicators: varying degrees of congestion, inflammatory infiltration, and edema, with each indicator also scored on a scale of 1–9 points [31].

### Inhibitory effect of pre-established biofilms on *C. albicans*

To each well of the 6-well plate, add 5 mL of SDB medium and inoculate with 0.1 mL of *C. albicans* suspension. Incubate at 37°C for 24 hours. After incubation, discard the medium and rinse the wells once with PBS (pH 7.2–7.4). Add 5 mL of fresh SDB medium with DMSO (10 μL), SR extract (20 μg/mL or 100 μg/mL), or CSR extract (20 μg/mL or 100 μg/mL) to the well and incubate for 24 hours, repeating the previous treatment. Fix the biofilm by adding 5 mL of 4% paraformaldehyde solution to each well and incubating for 15 minutes. Discard the paraformaldehyde and rinse once with PBS (pH 7.2–7.4). Stain the wells with 2 mL of 0.1% crystal violet aqueous solution for 15 minutes, then discard the staining solution and rinse once with PBS. Allow the plates to air dry. To quantify the biofilm, add 1 mL of 33% acetic acid to each well of the stained 6-well plate and allow it to stand for 15 minutes to ensure complete decolorization. Pipette 100 μL of the decolorized solution into a 96-well plate, and use a microplate reader to measure the absorbance at 550 nm [32]. In addition, biofilms were stained with fluorescent stains SYTO 9 and visualized under a confocal laser microscope (LSM 900, ZEISS) [33]. To quantify biofilm structures, COMSTAT software was used to determine biovolumes (μM3 μM2), and mean biofilm thicknesses (μM). Two independent cultures were performed and 6 wells per sample [34].

### Research on the inhibitory effect of hyphal phase on *C. albicans*

Prepare three sterile test tubes, each containing 5 mL of SDB and RPMI 1640 culture medium (1:1). Add 0.1 mL of newborn bovine serum to each tube, followed by 0.1 mL of a three-fold gradient-diluted *C. albicans* suspension. Subsequently, add 10 μL of DMSO, SR extract, or CSR extract to the respective tubes, ensuring a final drug concentration of 100 μg/mL. Seal the test tubes and incubate them in a constant temperature shaker at 37 °C with a speed of 150 rpm for 24 hours. Extract 1 mL of the fungal suspension, perform centrifugation at 12,000 rpm for 1 minute, and remove the supernatant. Following fixation in 2.5% glutaraldehyde (20 minute) and PBS washing, the samples underwent stepwise dehydration in increasing ethanol concentrations (30%, 50%, 70%, 90%, and absolute ethanol), with 15 minute per step. After samples were dried and coated with gold, the morphology of *C. albicans* was visualized using TESCAN MAIA 3 GMU SEM (Tescan, Brno, Czech Republic).

### The effect of different incubation times of CSR on biofilm and mycelial phase

To each well of a 6-well plate, add 5 mL of SDB and RPMI 1640 culture medium (1:1) along with 0.1 mL of newborn bovine serum, and inoculate with 0.1 mL of the *C. albicans* suspension; Add 10 μL of DMSO or CSR extract to achieve a terminal drug concentration of 100 μg/mL. After incubation for 18, 24, and 30 hours respectively, discard the culture medium and proceed with staining as described in above for subsequent observation.

### The effect of CSR on estrogen induced mycelial phase

To each well of a 6-well plate, add 5 mL of SDB and RPMI 1640 culture medium (1:1) along with 0.1 mL of estradiol solution at a concentration of 10 mg·mL$^{-1}$. Inoculate with 0.1 mL of *C. albicans* suspension, and add 10 μL of DMSO, SR extract, or CSR extract separately, ensuring a final drug concentration of 100 μg/mL in the system. After 24 hours of incubation, discard the culture medium and perform staining as described in above for subsequent observation.

### The effect of CSR on the expression of pathogenic genes of *C. albicans*

*C. albicans* hyphal culture and induction using newborn bovine serum followed the aforementioned protocol. For RNA extraction, fungal cells were collected after 24-hour treatment with CSR extract (100 μg/mL). Total RNA from *C. albicans* was isolated by means of combining liquid nitrogen grinding and an RNA extraction kit. A Nanodrop2000 spectrophotometer was employed to measure the concentration and purity of the RNA, and agarose gel electrophoresis was utilized to assess integrity. An Agilent 5300 Bioanalyzer was utilized to determine the RNA Integrity Number (RIN) For single-library

construction, the RNA sample requirements were that the total RNA content should be ≥ 1 µg and the concentration should be ≥ 30 ng·µL$^{-1}$, a RIN value >6.5, and an OD260/280 ratio between 1.8 and 2.2.

The Illumina platform was utilized to conduct eukaryotic mRNA sequencing, enabling comprehensive analysis of all mRNA derived from specific tissues or cells of eukaryotic organisms during a given stage. Library was prepared by means of the Illumina NovaSeq Reagent Kit. Sequencing was carried out Shanghai Meiji Biomedical Technology Co., Ltd. The abundance of transcripts serves as an indicator of gene expression levels, with higher transcript abundance correlating with increased gene expression. In RNA sequencing analysis, gene expression levels are determined based on the read count of sequences mapped to specific genomic regions. The quantitative analysis of gene and transcript expression levels was analyzed using software the RSEM software(http://deweylab.biostat.wisc.edu/rsem/), Version 1.3.3. This analysis facilitates the examination of gene/transcript expression variations across varied samples and aids in elucidating gene regulatory mechanisms by integrating functional sequence information. After obtaining the gene Read Counts, differential expression analysis was carried out to identify variations in gene expression across two or more samples. This process aimed to identify differentially expressed genes (DEGs) and explore their functional significance. Using DESeq2 software (Version 1.24.0, obtainable at http://bioconductor.org/packages/stats/bioc/DE

Seq2/), the analysis was executed. The standard thresholds for identifying significant DEGs were established with a false discovery rate (FDR) < 0.05 and |log2 fold change (FC)| ≥ 1.Genes meeting both of these conditions were considered differentially expressed.

Employing 18S rRNA as a housekeeping gene, we identified five crucial genes with differential expression for validation via reverse transcription quantitative polymerase chain reaction (RT qPCR) to ascertain the accuracy of the RNA-seq data. The precise sequences of the primers are detailed in Table 1. The qPCR mix, with a overall volume of 10 µL, contained 5 µL of 2X ChamQ SYBR Color qPCR Master Mix, 1 µL of cDNA template, 0.5 µL respectively for the forward and reverse primers at a final concentration of 5 µM, and 3 µL of RNase-free water. The PCR cycle conditions are outlined in Table 2. Following preparation, the samples-loaded 96-well plate was positioned in a fluorescence quantitative PCR instrument for amplification. The internal reference for gene expression was set as 18S rRNA, with uninduced *C. albicans* serving as the control group. The analysis of relative gene expression levels was carried out by the 2$^{-\Delta\Delta Ct}$ method.

## Comparison of components before and after stir frying of SR

Weigh precisely 5.0 mg of SR and CSR extract and employ 10 mL of chromatography-grade methanol for dissolving. Sonicate to make a 0.5 mg/mL stock solution, and subsequently centrifuge it at 12000 rpm for 10 minutes. After centrifugation,

**Table 1. Primer sequence.**

| Number | Gene name | Primer name | Primer sequence (5'→3') | Length (bp) |
|---|---|---|---|---|
| 1 | HWP1 | HWP1-F | GGTGCTATTACTATTCCGGAAT | 180 |
| | | HWP1-R | AATAATAGCAGCACCGAAAGTC | |
| 2 | CPH1 | CPH1-F | TGCAACACTATTTATACCTCCA | 214 |
| | | CPH1-R | ACCATTAGCACTCTGTAGTGAA | |
| 3 | ALS1 | ALS1-F | GGATCTGTTACTGGTGGAGCTGTTG | 148 |
| | | ALS1-R | ATGTGTTGGTTGAAGGTGAGGATGAG | |
| 4 | EFG1 | EFG1-F | TATGCCCCAGCAAACAACTG | 202 |
| | | EFG1-R | TTGTTGTCCTGCTGTCTGTC | |
| 5 | CSH1 | CSH1-F | CTGTCGGTACTATGAGATTG | 164 |
| | | CSH1-R | GATGAATAAACCCAACAACT | |
| Internal reference | 18SrRNA | 18SrRNA-F | GGGGATCGAAGATGATCAGA | 553 |
| | | 18SrRNA-R | CACGACGGAGTTTCACAAGA | |

**Table 2. PCR cycle conditions.**

| Process | Temperature | Time | Cycle conditions |
|---|---|---|---|
| Pre denaturation | 95°C | 5 min | 1 |
| Denaturation | 95°C | 5 sec | 40 |
| Annealing | 55°C | 30 sec | |
| Extend | 72°C | 40 sec | |

perform a gradient dilution to achieve a final concentration 500 μg·mL$^{-1}$ and examine for any precipitation. If precipitation occurs, continue to dilute with methanol until no further precipitation is observed. Record the volume of methanol used, which constitutes the test solution. Gallic acid, ellagic acid, catechins, methyl gallate, pyrogallic acid, and protocatechuic acid were selected as quality indicators. Prepare a control solution at a concentration of 200 μg/mL.

Each sample was examined in triplicate with the aid of a quadrupole mass spectrometer, specifically the Triple Quad 5500 model from ABSCIEX (located in Foster City, CA), and the operation of this spectrometer was performed by Analyst® 1.6 software. The mass spectrometer was directly connected to an ultra-high performance liquid chromatography (UHPLC) setup provided by Shimadzu Corporation (Kyoto, Japan). This UHPLC setup comprised a SIL-30 AC autosampler, a CBM-20A Lite controller, a DGU-20A5 degasser, a CTO-20A column oven, and a pair of LC-30AD pumps.

The UHPLC settings were as follows: The chromatographic separation was realized by an Agilent Extend-C18 column(100 mm × 2.1 mm, with a particle size of 1.8 μm). The mobile phase was a mixture of ultra-pure water containing 0.1% formic acid (solvent A) and acetonitrile (solvent B), operated at a flow rate of 0.2 mL per minute. Gradient eluting profile was set as: from 0 to 1 minute, increasing from 5% to 50% B; from 1 to 5 minutes, increasing from 50% to 95% B; holding at 95% B for 5–6 minutes; decreasing from 95% to 5% B between 6 and 7 minutes; and maintaining at 5% B from 7 to 8 minutes. The volume of sample injected was 1 μL, and the temperature of the column was maintained at 30 °C throughout the process.

The mass spectrometry (MS) parameters were configured as follows: Detection was carried out in negative ionization mode, employing an electrospray ionization (ESI) technique. The ion spray voltage was adjusted to −4,500 V, and the ion source temperature was maintained at 550 °C. The curtain gas (CUR), as well as the ion source gases 1 and 2 (GS1 and GS2), were all set at pressures of 35, 55, and 55 psi, respectively, for the analysis of all compounds. The quantification process was executed in multiple reaction monitoring (MRM) mode. Table 3 provided the parameters related to the compounds for the eight analytes. The entrance potential (EP) was set at −10.0 V and the collision exit potential (CXP) at −17.0 V.

## Validation of the pharmacological substance ellagic acid

To confirm ellagic acid's role in CSR's inhibition of hyphal and biofilm formation, we conducted the following experiment: fungal suspensions were inoculated into 12-well plates containing 1 mL SDB medium per well and incubated for 24 hours.

**Table 3. Compound related parameters of iconic components.**

| Chemical compound | Parent Ion | Secondary fragment ions | Declustering potential (kv) | Collision voltage (kv) | mode |
|---|---|---|---|---|---|
| Gallic acid | 169.1 | 125.0, 79.0 | −51.15, -45.45 | −10.06, -28.92 | − |
| Ellagic acid | 301.3 | 284.1, 145.1 | −45.38, -41.32 | −39.41, -49.01 | − |
| Catechin | 289.27 | 245.1, 203.0 | −51.33, -56.78 | −19.10, -25.28 | − |
| Methyl gallate | 183.15 | 124.1, 168.0 | −30.57, -40.56 | −18.33, -20.67 | − |
| Pyrogallic acid | 125.11 | 79.0, 69.1 | −56.04, -67.47 | −25.14, -24.78 | − |
| Protocatechuic acid | 153.12 | 109.0, 108.0 | −49.89, −51.26 | −25.97, -17.02 | − |

After medium removal, cells were resuspended in a 1:1 mixture of SDB and RPMI 1640 for hyphal induction. Test groups received either: (1) DMSO control, (2) CSR extract (100 µg/mL), or (3) CSR extract (100 µg/mL) plus ellagic acid (4 µg/mL). Following 24 hours incubation, hyphal formation was assessed by crystal violet staining (as described previously) and microscopic examination.

### Statistical analysis

Data analysis was conducted utilizing GraphPad Prism version 8, and the results of three replicates were presented as the mean ± standard deviation (SD). The assessment of statistical significance was carried out via one-way ANOVA. Statistical significance was deemed to be indicated by a *P value of below 0.05.

## Results

### The therapeutic effects of SR and CSR on mice with VVC

A mouse model of VVC was successfully established, following by a 7-day treatment as outlined in Fig 1A. On day 7, the results of vaginal lavage culture were presented in Fig 1B and C. *C. albicans* appeared as distinct white, circular colonies or spots, whereas other vaginal bacteria formed oily spots on SDA medium. The findings showed that the number of *C. albicans* colonies in the lavage fluid from CSR treatment group was significantly reduced compared to the model group, although residual *C. albicans* growth was still observed. This indicated that CSR exhibited inhibitory effects against *C. albicans*, albeit with lower efficacy compared to clotrimazole. Similarly, *in vitro* experiments confirmed CSR's inhibitory effect on *C. albicans* (S2 Table, S1 and S2 Figs).

Histopathological results (Fig 1D) revealed that the vaginal mucosa in the healthy group had a flat epithelium without keratinization, ulceration, or erosion, and the submucosal tissue showed no congestion, inflammatory infiltration, or edema. In contrast, the model group displayed variable keratinization thickness in the vaginal mucosal epithelium, significant inflammatory cell infiltration, localized erosion and necrosis, as well as mild submucosal congestion. After SR treatment, the mucosal epithelium exhibited upper-layer keratinization, and the submucosal tissue showed mild congestion with inflammatory cell infiltration. In comparison, the CSR group showed no significant keratinization or congestion in the submucosal tissue and marked improvement in inflammation. Notably, a large number of hyphae were observed in the pathological sections of the model group. Even after clotrimazole treatment, some hyphae remained attached to the vaginal wall, whereas they were almost entirely absent following CSR administration. Quantitative analysis of pathological damage further confirmed that CSR reduced the pathological damage to vaginal tissue and mucosa (Fig 1E and F). The pathogenicity of *C. albicans* largely stem from its ability to form biofilm formation and transition to the hyphal phase, both of which are central to the pathogenesis of VVC [35]. These findings suggested that CSR may effectively inhibit biofilm formation and hyphal transition, thereby reducing *C. albicans* adhesion to the vaginal wall.

### The effects of SR and CSR on the biofilm and mycelial phase of *C. albicans*

To assess the potential anti-biofilm activity of SR and CSR, the biomass of *C. albicans* biofilms was quantified using a crystal violet assay. As shown in Fig 2A, doses of 20 and 100 µg/mL significantly reduced biofilm formation, with CSR showing a more pronounced inhibitory effect than SR. CLSM images revealed robust biofilm formation in the control group, whereas CSR treatment resulted in diminished biofilm attachment and densities (Fig 2B). COMSTAT analysis indicated that both SR and CSR significantly reduced biofilm biomass by 77.1% and 56.7%, respectively, as well as average thickness by 89% and 81.1%, respectively, when compared to the control group (Fig 2C).

Serum induction, a method that rapidly transitions *C. albicans* from the yeast phase to the hyphal phase, revealed distinct differences between CSR and SR. After 24 hours co-incubation with CSR, hyphae were nearly absent, and only the yeast form remained, whereas SR-treated samples retained abundant hyphae (Fig 2D). The temporal effects of CSR

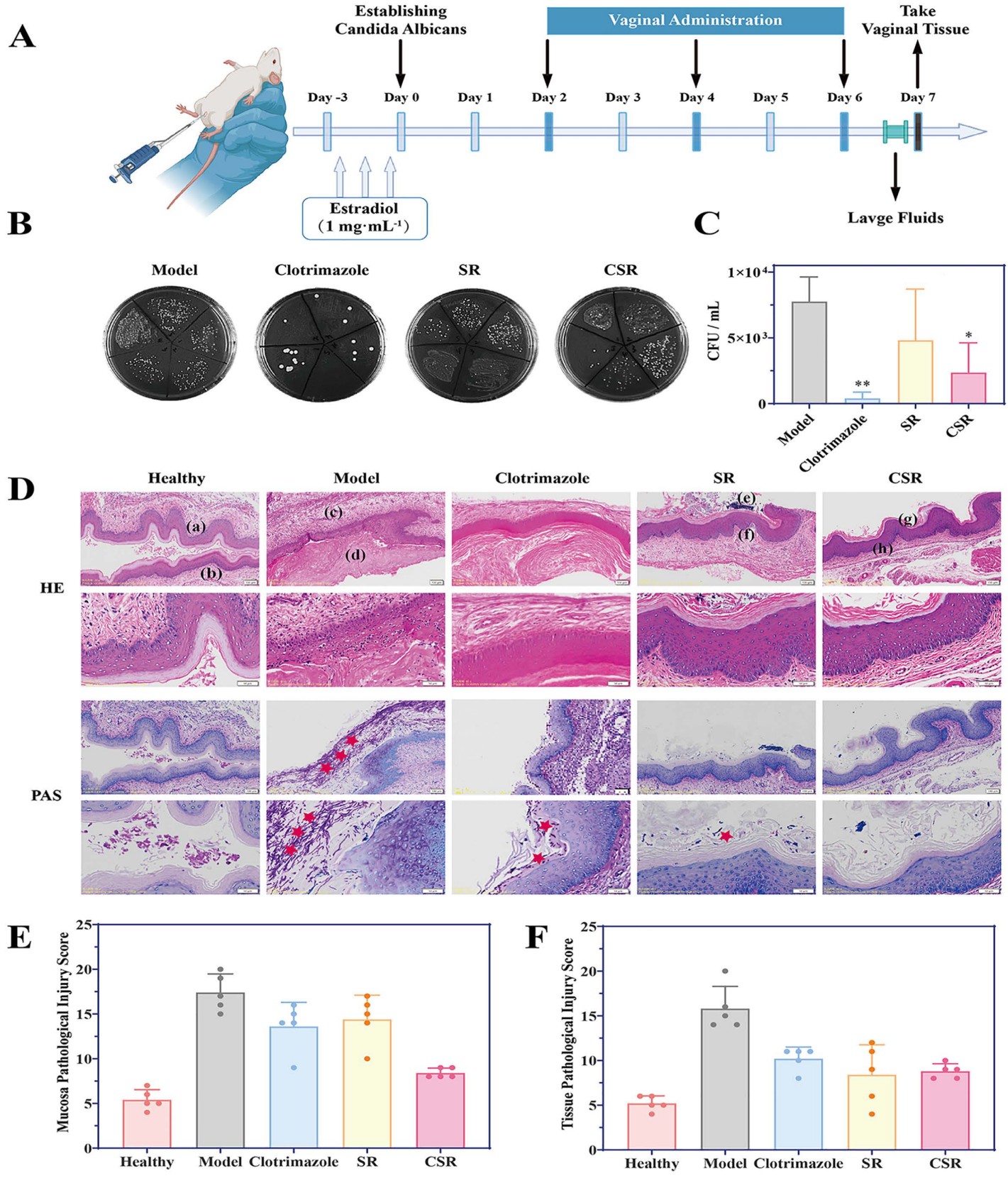

**Fig 1. Study on the efficacy of SR extract and CSR extract in treating VVC. (A)** Construction and administration of a mouse model of VVC. **(B-C)** Cultivation of *C. albicans* in vaginal a lavage fluid of mice after 7 days of treatment. Results are shown as means ± SD. **$P < 0.01$, *$P < 0.05$ compared to the model group. All groups n = 5. **(D)** HE staining and PAS staining of mouse vaginal tissue. **(a)** Normal vaginal mucosa; **(b)** Normal vaginal submucosal tissue; **(c)** Inflammatory vaginal submucosal tissue; **(d)** Necrotic shedding of vaginal mucosal cell clusters; **(e)** Damaged vaginal mucosa; **(f)** Vaginal submucosal tissue with mild inflammation; **(g)** Basically normal vaginal mucosa; **(h)** Normal vaginal submucosal tissue. ★ indicates *C. albicans* attachment to the vaginal surface. **(E-F)** Evaluation of pathological damage to vaginal mucosa and submucosal tissue in mice.

on serum-induced *C. albicans* over 18, 24, and 30 hours were shown in Fig 2E. Compared to the control, CSR treatment significantly reduced biofilm biomass and inhibited mycelial development. Microscopic analysis revealed progressive hyphae growth in untreated cultures over time, whereases CSR-treated cultures showed a marked reduction in hyphal formation. In the CSR treatment group, at 18 hours, only yeast cells were present. By 24 hours, shorter and less abundant hyphae appeared, indicating impaired invasive growth. Notably, the mature biofilm stage of CSR-treated *C. albicans* contrasted with the cell dispersion stage in untreated samples, with higher biofilm biomass in the former. Biofilm formation by *C. albicans* progresses through four stages: (1) yeast cell adhesion (adherence), (2) proliferation to form a basal layer (initiation), (3) maturation with hyphal growth and extracellular matrix production (maturation), and (4) yeast cell dispersal to colonize new surfaces (disperal) [9]. The reduced biofilm in the control group may reflect biomass detachment during the cell dispersion stage. In contrast, CSR delayed biofilm cycling and suppressed hyphal transformation, ultimately reducing biofilm biomass and cell counts after 30 hours.

Estrogen, a weaker inducer of hyphal formation compared to serum, interacts with *C. albicans* estrogen receptors to enhance adhesion to vaginal epithelial cells, thereby increasing infection risk. As shown in Fig 2F, CSR significantly inhibited biofilm and hyphal formation induced by estrogen at doses of 100 and 500 µg/mL. CSR-treated samples, *C. albicans* colonies appeared small and scattered, in stark contrasts to the untreated samples, which exhibited densely agglomerated hyphal colonies. These findings confirmed that CSR was more effective than SR in inhibiting biofilm and hyphae formation in *C. albicans*.

### The effect of CSR on the expression of biofilm-related genes in *C. albicans*

To investigate the impact of CSR on the gene expression profile of *C. albicans*, DESeq2 analysis was performed on transcriptomic samples to identify differentially expressed genes (DEGs). As illustrated in Fig 3A and B, co-incubation with CSR induced significant transcriptional changes in serum-induced *C. albicans*, resulting in 2248 DEGs, which included 966 upregulated and 1282 downregulated genes. Further analysis focused on virulence genes associated with *C. albicans*, as shown in Fig 3C. Genes involed in biofilm regulation, such as ALS1 and ALS3, were part of the agglutinin-like sequence family [36]. Following CSR treatment, ALS1 expression was significantly downregulated, while ALS3 showed a slight upregulation. Additionally, HWP1 and BCR1 encode hyphal wall protein-related genes [37]. CPH1, a transcription factor involved in the hyphal phase, showed changes consistent with the inhibition of hyphal development [38]. Similarly, the hyphal growth-related gene EFG1 [39]was marked downregulation post-treatment, further supporting the inhibition of hyphal development. Similarly, genes associated with biofilm induction, such as TUP1, a hyphae repressor factor [32], were upregulated following CSR treatment, aligning with its suppressive effect on cell dispersion stage. Conversely, biofilm-repressed genes, including CSH1, RAS1, and SMI1 [40], were significantly downregulated after CSR incubation, further corroborating its anti-biofilm activity. To validate the transcriptomic data, RT-qPCR analysis was conducted using 18S rRNA as an internal reference gene. Five pathogenicity-related genes were selected for verification, and gene expression levels were standardized and log-transformed (base 2). The RT-qPCR results (Fig 3D) demonstrated trends consistent with the transcriptome analysis, confirming the reliability of the data. These findings provided genetic evidence supporting the inhibitory effects of CSR on biofilm formation and hyphal development in *C. albicans*.

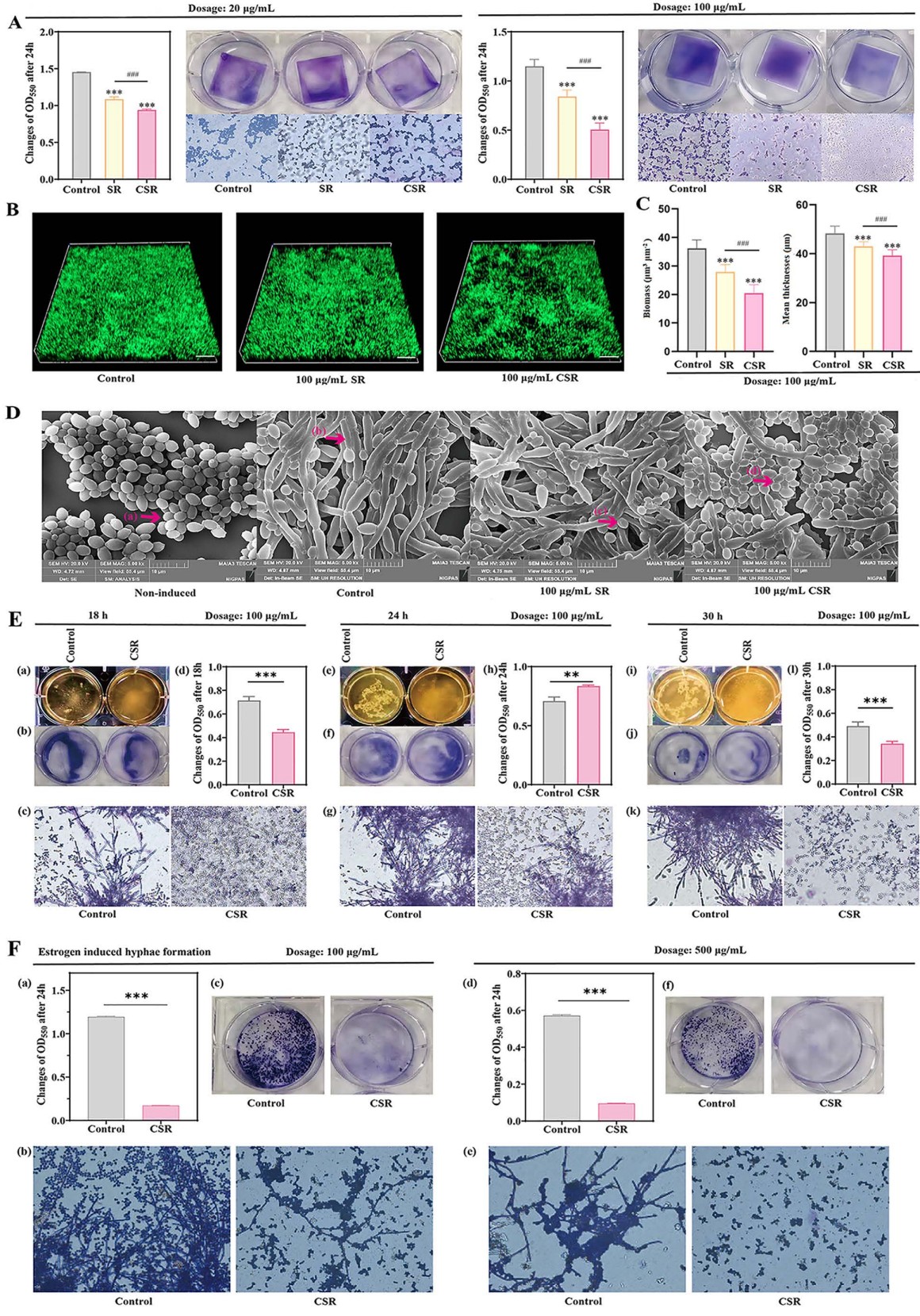

**Fig 2. Inhibitory effect of SR extract and CSR extract on biofilm and hyphal formation of *C. albicans*. (A)** Inhibition of pre-established biofilms formed by *C. albicans*. Results are shown as means ± SD. \*\*\**P* < 0.001 compared to the control group. ###*P* < 0.001 compared with the SR extract group. All groups n = 3. The magnification of the optical microscope is 400X. **(B)** Confocal laser scanning microscopy images of *C. albicans* pre-established biofilms treated with SR extract (100 μg/mL) and CSR extract (100 μg/mL) for a duration of 24 hours. Scale bars represent 50 μm. **(C)** Biofilm alterations were quantified utilizing COMSTAT. Two independent experiments were conducted (six wells per sample). \*\*\**P* < 0.001 compared to the control group. ###*P* < 0.001 compared with the SR extract group. **(D)** Changes in the dimorphism of *C. albicans* hyphae induced by SR extract (100 μg/mL) and CSR extract (100 μg/mL). Electron microscope image, scale bars represent 10 μm. **(a)** *C. albicans* yeast phase; **(b)** *C. albicans* hyphal phase; **(c)** Undeveloped hyphal phase; **(d)** Wrinkled mycelial phase. **(E)** The effect of different incubation times of CSR extract (100 μg/mL) on the biofilm and hyphal phase of *C. albicans*. Results are shown as means ± SD. \*\*\**P* < 0.001 compared to the control group. All groups n = 3. The magnification of the optical microscope is 400X. **(F)** The effect of CSR extract (100 μg/mL and 500 μg/mL) on estrogen-induced hyphal phase. The magnification of the optical microscope is 400X. Results are shown as means ± SD. \*\*\**P* < 0.001 compared to the control group. All groups n = 3. All control groups contained DMSO without extract.

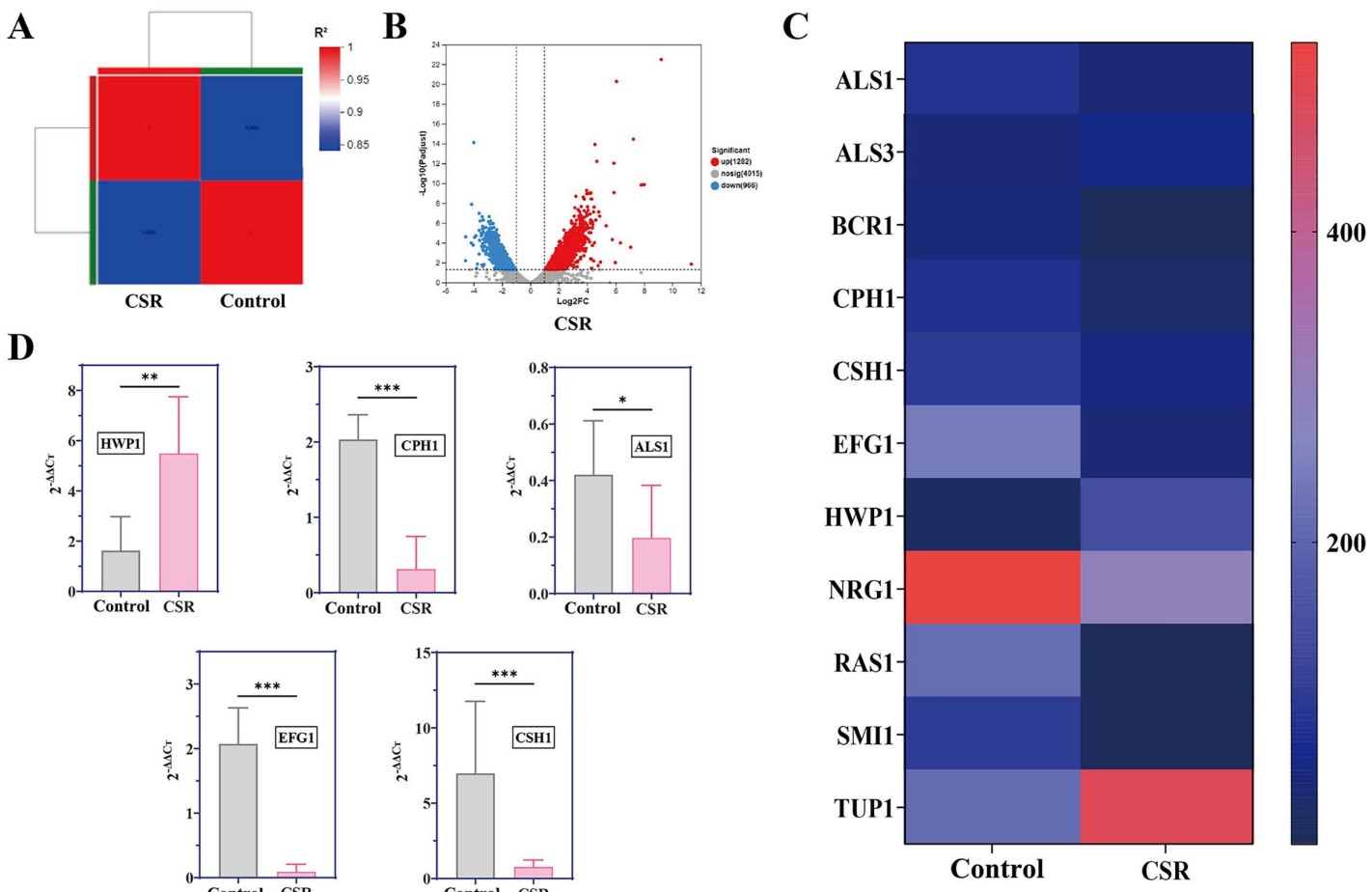

**Fig 3. Study on the expression of mRNA related to biofilm formation of *C. albicans* by CSR extract (100 μg/mL). (A)** Differentially expressed genes (DEGs) analysis performed on transcriptomic samples. **(B)** Differences in gene expression levels of *C. albicans* after incubation with CSR. **(C)** Differential expression of adhesion-related genes in *C. albicans* after incubation with CSR extract. Red represents high gene expression, blue represents decreased gene expression. **(D)** RT-qPCR validation results. Results are shown as means ± SD. \*\*\**P* < 0.001, \*\**P* < 0.01, \**P* < 0.05 compared to the control group. All groups n = 3. The control group contained DMSO without extract.

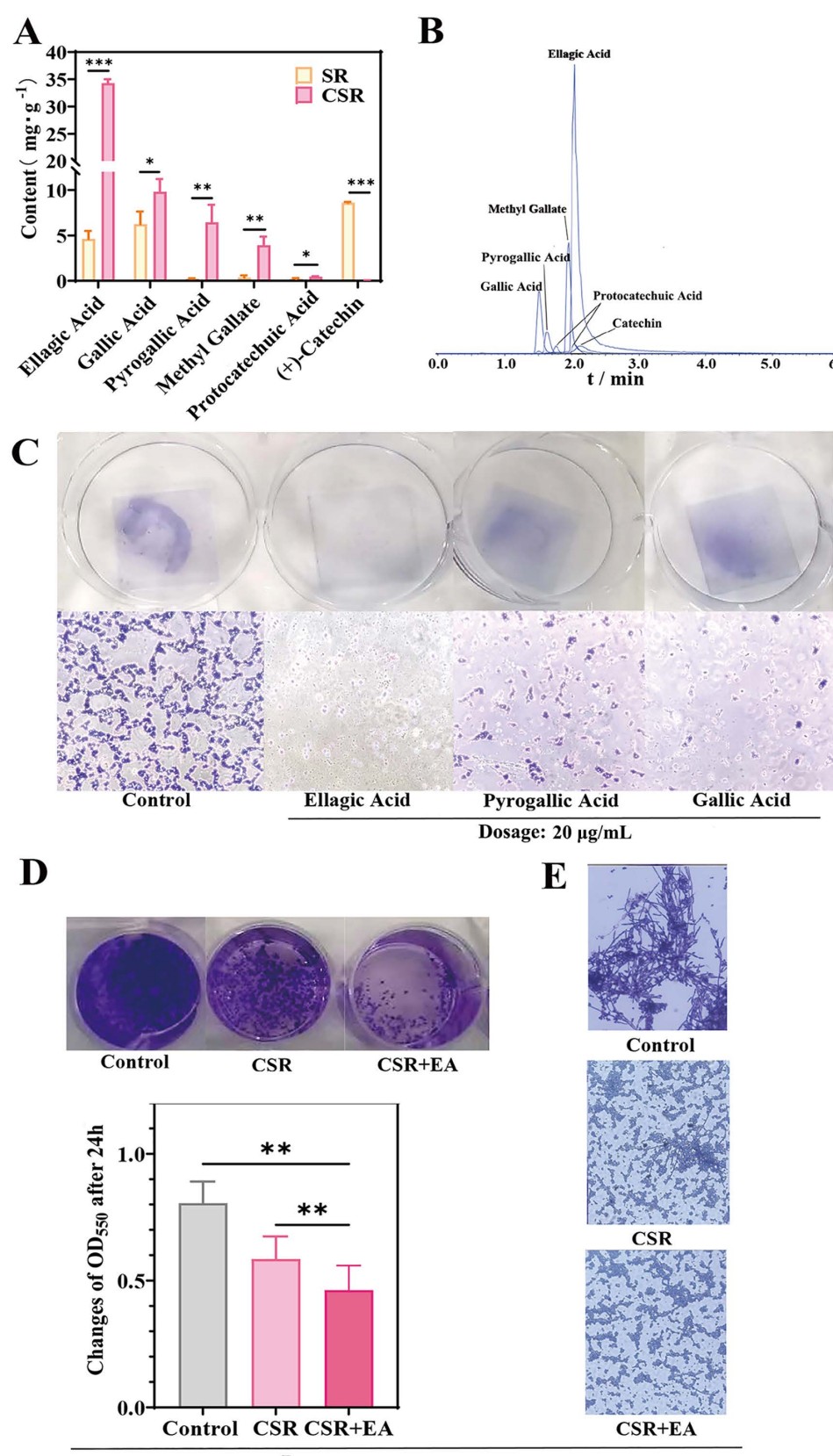

**Fig 4. Study on the pharmacological substances of CSR extract in inhibiting biofilm formation and hyphal phase. (A)** Contents of main components in SR extract and CSR extract. Results are shown as means±SD. ***$P < 0.001$, **$P < 0.01$, *$P < 0.05$ compared to the SR group. All groups n = 3. **(B)** Chromatograms of respective compounds in positive and negative modes. **(C)** Inhibition of *C. albicans* biofilm by monomeric components. The concentrations of Ellagic acid, Pyrogallol and Gallic acid are all 20 μg/mL. **(D)** Inhibition biofilm of *C. albicans*. The concentration of Ellagic acid is 4 μg/mL and CSR extract is 100 μg/mL. Results are shown as means±SD. **$P < 0.01$ compared to the control group. All groups n = 3. **(E)** Inhibition hyphal phase of *C. albicans*. The concentration of Ellagic acid is 4 μg/mL and CSR extract is 100 μg/mL. All control groups contained DMSO without extract.

## Investigation of medicinal substances in CSR

To explore the chemical changes induced by charcoal frying and their potential contributions to the distinct effects on *C. albicans* biofilm and hyphal stages, we conducted precise quantification of key components in SR and CSR extracts (S2 Table, S3 Fig). As shown in Fig 4A, the most abundant components in SR were catechin, gallic acid (GA), and ellagic acid (EA), while CSR was predominantly enriched in EA, GA, and pyrogallic acid (PYG). Chromatograms, mass spectra, and structural formulas of these components were provided in Fig 4B and S4 Fig. Remarkably, the content of EA, GA and PYG increased significantly by 7.44-fold, 1.57-fold and 28.09-fold respectively, while catechin levels markedly decreased after charcoal frying.

We further compared the *in vitro* inhibitory effects of three major monomeric components of CSR on biofilm formation. As shown in Fig 4C, EA exhibited biofilm inhibitory effect. To further validate EA's contribution to CSR's antifungal efficacy, an equivalent amounts of EA were added to the CSR extract. As a result, the biofilm and hyphal structures of *C. albicans* were almost completely eliminated in CSR+EA group, as evidenced by the disappearance of crystal violet precipitate and hyphal structures (Fig 4D and E). These findings confirmed that EA was the key bioactive compound in CSR responsible for its enhanced inhibitory activity on *C. albicans* biofilm and hyphal development.

## Discussion

The pronounced inhibitory effect of SR on *C. albicans* has been well-documented previously [41], and our findings presented in S1 Fig also showed similar effects. Our study here comprehensively examined a novel direction, which is the CSR's ability to inhibit yeast-to-hypha differentiation and biofilm formation in *C. albicans*. Following CSR treatment, the nearly complete elimination of hyphal *C. albicans* adhering to the vaginal mucosa in the VVC model mice was observed. CSR's remarkable inhibitory effects on biofilm formation and hyphal growth were further corroborated through quantitative biofilm analysis, morphological observations, and gene expression studies conducted *in vitro*. In contrast, persistent hyphal formation was observed in the SR treatment groups both *in vitro* and *in vivo* experiments. These findings highlighted the critical role of carbonization in enhancing the inhibition of hyphal formation in *C. albicans*.

To elucidate the molecular basis of CSR's anti-biofilm activity against *C. albicans*, we assessed the expression patterns of key genes involved in cell adhesion and filamentation during biofilm development. Following CSR treatment, the expressions of ALS1, BCR1, CPH1, CSH1, EFG1, NRG1, RAS1, and SMI1 expression were downregulated, while ALS3, HWP1 and TUP1 were upregulated. The RAS1-cAMP-EFG1 signaling pathway controls yeast-to-hypha conversion, which promotes biofilm formation and facilitates host tissue invasion [42]. Previous studies showed that by targeting the Ras1-cAMP-Efg1 regulatory system, certain natural products effectively impair multiple virulence determinants in *C. albicans*, including surface attachment, hyphal growth, and biofilm assembly [43]. Ras is known to play a pivotal role in orchestrating the morphogenetic transitions between yeast and hyphal forms, as well as in mediating cell adhesion and biofilm development in *C. albicans*. Moreover, EFG1 emerges as the principal regulator orchestrating biofilm formation. In our study, Ras1-cAMP-Efg1 pathway-related genes (EFG1 and RAS1) were also found significantly down-regulated. The down-regulated genes may contribute to the anti-adhesive and anti-yeast-to-hyphae transition of CSR. In addition, Tup1, a transcriptional repressor, was upregulated after CSR treatment. The transcriptional regulator Tup1 suppresses hyphal development in *C. albicans*. When TUP1 was deleted in strain CAI4, the cells grew exclusively in filamentous form [44].

These results implicated that CSR might impede the initial adhesion of *C. albicans* cells and yeast-tohypha switch, thus prevent biofilm formation.

Conventional antifungal agents, including azoles (e.g., fluconazole), polyenes (e.g., amphotericin B), and echinocandins (e.g., caspofungin), primarily target the fungal cell wall or cell membrane to exert fungistatic or fungicidal effects. However, prolonged clinical use of single-target antifungal therapies has led to a significant prevalence of recurrent infections, with many cases exhibiting azole resistance [45].Biofilm formation is a major factor contributing to the resistance of *C. albicans* to traditional antifungal agents. Notably, several natural products and small-molecule compounds have demonstrated promising activity in inhibiting the biofilm formation of *C. albicans* [46]. In the present study, the observed differences efficacy in biofilm inhibition between SR and CSR is believed to attribute to the compositional changes induced by carbonization. Tannins constitute the primary active components transformed during SR carbonization [47]. While total tannin content remained stable after processing, we observed significant structural conversions between hydrolyzable and condensed tannins. Carbonization-induced cleavage of glycosidic/ester bonds in hydrolyzable tannins liberated free GA, which subsequently converted to PYG through thermal degradation, ultimately increasing PYG concentration by 15-fold. Simultaneously, macromolecular ellagitannins were hydrolyzed into EA derivatives [47], collectively explaining the increases in EA, GA and PYG content. In our study, EA showed inhibitory effect against *C. albicans* biofilms, and this observation was consistent with findings reported in previous study [48]. Notably, while prior studies identified PYG as the most potent inhibitor (sessile $MIC_{50} = 40$ µg/mL) [49,50], its low concentration in CSR (0.64%) suggests EA contributes more significantly to CSR's overall antifungal effects due to its higher abundance (3.4%).

In summary, EA is identified as the primary bioactive compound responsible for the anti-biofilm activity of CSR, due to its abundant content and significant effect. On the other hand, the relatively low activity of SR may be attributed to its composition. GA and catechin are rich in SR. However, both GA and catechin have limited antibiofilm activity [50]. Therefore, carbonization technology is crucial for the clinical application of SR, a factor often overlooked by clinicians.

The increasing resistance of Candida species to antifungal drugs in recent years has led to a shortage of effective treatments. As a valuable scientific and technological resource with unique advantages in China, TCM offers high efficacy, broad-spectrum activity, and low toxicity strategy, and has been widely applied in clinical settings [51]. TCM herbs can intervene in multiple pathways to inhibit the formation and maintenance of *C. albicans* biofilms. Herbal formulas such as Huanglian jiedu decoction and Longdan xiegan decoction have been shown to affect the transition of *C. albicans* from the yeast phase to the hyphal phase [52–54]. Compound like tetrandrine also influences this transformation [55]. Additionally, components such as andrographolide and paeoniflorin [56,57] have been reported to disrupt mature biofilms. Magnolia polyphenols have been shown to impede the conversion from the yeast phase to the hyphal phase, disrupt mature biofilms, and inhibit adhesion [58,59]. This study is the first to propose the concept that carbonization technology enables SR to regulate the yeast-to-hypha transition and biofilm formation in *C. albicans* by enhancing the content of EA and PYG. The hallmark of TCM lies in the synergistic action of its various components. The multiple components of CSR extract may work together in a combined or synergistic manner to effectively inhibit biofilm formation and suppress hyphal growth of *C. albicans*. Furthermore, although CSR was not as effective as clotrimazole in reducing the total count of *C. albicans*, it significantly diminished biofilm formation and hyphal growth, thereby alleviating the adhesion and invasion of the microbial community to the vaginal wall. This mechanism explains the disappearance of hyphal forms from the vaginal wall in the CSR-treated group, ultimately providing protective effects against vaginal damage, while SR did not achieve the same level of efficacy. In addition to the *C. albicans* studied in this research, there have been reports indicating that CSR exhibits antimicrobial activity against *Staphylococcus aureus* and *Trichophyton rubrum*. Additionally, EA demonstrates efficacy against *Helicobacter pylori* [60], *Trichophyton rubrum* [61], and *Candida auris* [62].

This study has several limitations that warrant discussion. First, while EA represents the most abundant bioactive component in CSR (3.4% of total composition), batch-to-batch variability in EA content (3.3–7.1% across twelve tested

batches) underscores the need for improved standardization protocols to ensure consistent therapeutic effects. Second, although our transcriptomic analysis revealed significant gene expression changes associated with biofilm inhibition, future studies should employ genetic manipulation approaches (e.g., knockout/overexpression strains) to establish causal relationships between these molecular changes and the observed antifungal activity. These mechanistic validations would strengthen the clinical translation potential of CSR as an antifungal agent.

## Conclusion

This study demonstrates that CSR extract inhibits *C. albicans* biofilm formation, suppresses hyphal growth, and modulates the expression of biofilm-related genes, indicating its potential to target key virulence factors and help overcome drug resistance. EA was identified as the primary active component of CSR. Taken together, CSR and EA show promise as alternative therapeutic agents for *C. albicans* infections, particularly those involving biofilms, which are difficult to treat. Future investigation and evaluation of CSR and EA may lead to novel antifungal strategies for VVC.

## Supporting information

**S1 Fig. Comparison of the inhibitory effects of SR extract and CSR extract on *C. albicans*.** (A) Validation of the antifungal effects of SR extract and CSR extract on *C. albicans* at concentrations of 20, 100 and 500 µg/mL. CSR exhibited superior inhibitory activity compared to SR at concentrations of 100 and 500 µg/mL. Compared with the control group, *** $P < 0.001$, ** $P < 0.01$, * $P < 0.05$. (B) Verification of growth status of *C. albicans* of CSR extract at concentrations of 100 µg/mL in SDA culture medium. CSR showed significantly stronger inhibitory effects against *C. albicans* at 100 µg/mL than SR. (C) The effects of SR extract and CSR extract on the microscopic morphology of *C. albicans* in 100 µg/mL. The cell membrane of *C. albicans* in the control group was intact, smooth, and well-defined. In the SR-treated group, mucus-like substances were observed between *C. albicans* cells, but no significant structural changes in the fungal cells were noted. In contrast, the CSR-treated group exhibited substantial cell wrinkling and deformation, with approximately half of the fungal cells rupturing and showing cytoplasmic leakage. All groups n = 3. All control groups contained DMSO without extract.
(DOCX)

**S2 Fig. Evaluation of the antifungal mechanism of CSR extract against *C. albicans*.** (A) The effect of CSR extract (100 µg/mL) on the cell membrane of *C. albicans*. Ergosterol, the primary component of the *C. albicans* cell membrane, was found to decrease in content after incubation with CSR. Compared with the control group, *** $P < 0.001$, ** $P < 0.01$. (B) Cell permeability of *C. albicans* after incubation with CSR extract (100 µg/mL). A marked increase in protein content in the supernatant of *C. albicans* incubated with CSR following vortexing. Compared with the control group, *** $P < 0.001$. (C) The effect of CSR extract (100 µg/mL) on the cell wall of *C. albicans*. The concentrations of fluconazole and caspofungin used as positive and negative controls were 10 µg/mL and 1 µg/mL. The main component of the *C. albicans* cell wall is β-1,3-glucan, which can be protected by sorbitol. When sorbitol was added to the culture medium, CSR continued to exhibit inhibitory effects. This indicated that CSR does not affect the synthesis of the *C. albicans* cell wall. Compared with the control group, ** $P < 0.01$. (D) Comparison of the inhibitory effect of CSR extract (100 µg/mL) on *C. albicans* in SDB medium with different pH values. The inhibitory activity of CSR against *C. albicans* was dependent on pH levels. All groups n = 3. All control groups contained DMSO without extract.
(DOCX)

**S3 Fig. Comparison of physical parameters and chemical composition between SR and CSR.** (A) Pieces, sections, and powder diagrams of SR and CSR. The surface, cross-section, and powder of SR are uniformly yellow, whereas those of CSR are uniformly black, with some pieces appearing ashed. (B) The content of total tannins, condensed tannins, and hydrolysable tannins in SR and CSR. Condensed tannins were undetectable after charcoal frying. (C) The difference

between ellagitannin and gallotannin in hydrolyzed tannins. Hydrolyzed tannins in both SR and CSR predominantly consisted of ellagitannins and gallotannins. Charcoal frying induced notable compositional changes, with a significant increase in ellagitannin content observed in CSR. Compared with the SR group, *** $P < 0.001$. All groups n = 3.
(DOCX)

**S4 Fig. Mass spectrometry information of compounds in CSR extract.** (A) Ellagic acid. (B) Gallic acid. (C) Pyrogallic acid. (D) Methyl Gallate. (E) Protocatechuic Acid. (F) Catechin.
(DOCX)

**S5 Fig. Study on the efficacy of fluconazole in treating VVC.** HE staining (A) and PAS staining (B) of mouse vaginal tissue. After fluconazole treatment, there was no significant keratinization or congestion in the submucosal tissue and marked improvement in inflammation. However, some hyphae remained attached to the vaginal wall.
(DOCX)

**S1 Table. The ancient Chinese documents records the treatment of fungal vaginitis with SR.**
(DOCX)

**S2 Table. MICs of SR extract, CSR extract and related components against *C. albicans*.**
(DOCX)

**S3 Table. Comparison of Physical Parameters Related to SR and CSR.**
(DOCX)

## Author contributions

**Funding acquisition:** Jinyun Song, Huangjin Tong, Wei Gu, Hongyu Zhao.

**Investigation:** Jinyun Song, Hongdan Wu, Bohui Song, Ruixiao Ma, Jinghan Gao.

**Writing – original draft:** Xuxi Cheng, Qinglian Hu.

**Writing – review & editing:** Yiwei Wang, Wei Gu, Hongyu Zhao.

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
