## [Decision Letter · Decision Letter 0]

14 May 2025

Dear Dr. Zhao,

Thank you for submitting your manuscript to PLOS ONE. After careful consideration, we feel that it has merit but does not fully meet PLOS ONE’s publication criteria as it currently stands. Therefore, we invite you to submit a revised version of the manuscript that addresses the points raised during the review process.

We look forward to receiving your revised manuscript.

Kind regards,

Satish kumar Rajasekharan

Academic Editor

PLOS ONE

2. ‘Please include your tables as part of your main manuscript and remove the individual files. Please note that supplementary tables (should remain/ be uploaded) as separate "supporting information" files.

 [This work was financially supported by Nanjing Health Science and Technology Development Fund Medical Key Technology Development Project (Grant No. ZKX22039), the First Phase Reserve Talent Project of The Second Hospital of Nanjing (Grant No. HBRCYL09), Medical Research Project of Jiangsu Province Health Commission in 2023 (H2023084)�Advanced Training Program for Leading Personnel in Traditional Chinese Medicine in Jiangsu Province(Jiangsu Traditional Chinese Medicine Science and Education [2022] no.17)�Zeng Bailin Esteemed Veteran Pharmacist Heritage Workshop of Jiangsu Province (Jiangsu Traditional Chinese Medicine Research and Education [2024] No. 4)�Jiangsu Pharmaceutical Association Jin Peiying Fund Project (Grant No. J2021002), the Natural Science Foundation of Jiangsu Province (BK20231308), National Key Research and Development Program of China ‘Research on intelligent recognition and production control technology for stir frying traditional Chinese medicine slices’ (Grant No. 2023YFC3504200) and National Famous Traditional Chinese Medicine Expert Inheritance Studio Construction Project (Grant No. State Administration of Traditional Chinese Medicine [2022]75).].

4. In the online submission form, you indicated that [All data generated or analyzed during this study are included in the article and other related information are available from the corresponding author on reasonable request.].

Additional Editor Comments (if provided):

Reviewers' comments:

Reviewer's Responses to Questions

**Comments to the Author**

1. Is the manuscript technically sound, and do the data support the conclusions?

Reviewer #1: Yes

Reviewer #2: Yes

Reviewer #3: No

Reviewer #4: Partly

2. Has the statistical analysis been performed appropriately and rigorously?

Reviewer #1: Yes

Reviewer #2: Yes

Reviewer #3: No

Reviewer #4: No

3. Have the authors made all data underlying the findings in their manuscript fully available?

Reviewer #1: Yes

Reviewer #2: Yes

Reviewer #3: No

Reviewer #4: Yes

4. Is the manuscript presented in an intelligible fashion and written in standard English?

Reviewer #1: Yes

Reviewer #2: Yes

Reviewer #3: No

Reviewer #4: No

Reviewer #1: The research has been conducted well with sound methodology and interesting results.

However, the 'Conclusion' is unsatisfactory and I suggest to re-write it - conclusion should be based on the findings of your research. Avoid general comments and be specific on what you conclude from your results. Restate your main findings, may add future directions to improve the knowledge. Further, do not include references in the conclusion.

Reviewer #2: The manuscript entitled “Carbonizing Technology Enables Sanguisorbae Radix to Inhibit Yeast-to-Hypha Differentiation and Biofilm Formation in Candida albicans” provides a detailed account of the ability of charred Sanguisorbae Radix (CSR) to inhibit the yeast-to-hypha transition and biofilm formation in C. albicans. Although the manuscript is impressive, several minor suggestions and corrections are necessary to improve clarity and scientific rigor. Based on the following points, I recommend major revision:

1.In the Methodology section titled “Research on the inhibitory effect of biofilms on C. albicans*,” the authors should clearly mention the concentration of the compound added during the assay.

2.In the same section, it appears that the assay was performed on pre-formed biofilms of C. albicans. If this is the case, the authors should revise the title of the methodology to reflect that the study investigates the effect on pre-established biofilms, not the initial biofilm formation.

3.The authors should clarify the impact of SR and CSR on the viability of C. albicans. This will help differentiate between anti-biofilm effects and possible antifungal (fungicidal or fungistatic) activity.

4.Although the authors have explained the results of the transcriptomic analysis well, a brief summary of the major transcriptomic findings should be incorporated into the Discussion section to strengthen the overall interpretation of the data.

5.In the methodology section titled “The effect of CSR on the expression of pathogenic genes of C. albicans*,” the authors should specify the concentration of CSR used, the time point of treatment, and the duration of incubation. This information is essential, especially since biofilm assays were conducted at various time points.

6.The authors are advised to italicize the terms “in vitro” and “in vivo” throughout the manuscript, in accordance with standard scientific formatting.

7.The authors should discuss the possible mechanism underlying the enhanced content of ellagic acid (EA), gallic acid (GR), and pyrogallic acid (PYG) in SR following carbonization to produce CSR.

8.In the sentence “To further validate EA’s contribution to CSR’s antifungal efficacy, an equivalent amount of EA was added to the CSR extract,” it should be clarified whether similar experiments were conducted with GR and PYG. If so, the authors should describe the impact of adding these compounds to CSR and explain how the results led to the conclusion that EA exhibits the strongest antifungal activity.

9.Additionally, the authors should state the exact concentration of EA used when added to CSR (i.e., the equivalent amount mentioned in the experiments).

10.In Figure 4D, there does not appear to be a significant visual difference in the microscopic images of the biofilms treated with CSR versus CSR+EA. To validate these findings, the authors are encouraged to repeat the experiment or include quantitative evidence to support the visual data.

Reviewer #3: Authors have explored an interesting topic but data presented in this study is either missing, questionable or irreproducible. Poor scientific writing, inappropriate methodology, results, discussion and graphical representation of the data. Hence, the study is unsuitable for the publication. Authors need to rethink the motive behind this study and rewrite, re-analyze, re-perform experiments.

Few comments have been mentioned as a comment in attached PDF file.

Reviewer #4: This manuscript presents a study investigating the enhanced antifungal properties of Charred Sanguisorbae Radix (CSR) against Candida albicans, focusing on yeast-to-hypha differentiation and biofilm formation. While the research is based on an interesting traditional medicine approach, it suffers from several methodological and interpretational weaknesses that limit its scientific rigor and clinical relevance. Some concerns pertaining to the study are listed below:

•While clotrimazole is used as a control, the manuscript does not sufficiently compare CSR’s efficacy with clinically relevant concentrations of standard antifungals (e.g., fluconazole or echinocandins. Also, clotrimazole usage as positive control is not consistent across experiments.

•Quantitative metrics (MICs, biofilm IC50 and inhibition percentages) are missing and phrases like strong inhibition were used. Quantitative representations enhance reader comprehension on the efficacy of CSR.

•Although CSR inhibits hyphal transition, it fails in fungal burden elimination especially in comparison to clotrimazole which is a serious drawback. How do the authors justify clinical relevance in this case?

•The study lacks pharmacokinetic or toxicity studies of CSR or EA in vivo, which limits claims about therapeutic potential. Cytotoxicity data of CSR on mammalian cells are not provided, raising safety concerns for clinical application. Also, charring and carbonizing can produce carcinogens which can be detrimental to clinical application. The authors should address these issues.

•The justification for using CSR based on traditional Chinese medicine is not well-supported by modern pharmacological rationale. Some references are outdated or rely heavily on non-peer-reviewed Chinese sources. Also, some claims and statements pertaining to TCM are missing in the introduction.

•While gene expression changes are reported, causal links between these gene alterations and CSR’s effects are assumed but not experimentally validated. The role of CSR and EA vs PYG in modulating specific pathways is not dissected. A more holistic discussion on the interplay of gene expression changes and how CSR modulates these metabolic pathways to possibly interpret a mode of action should be done.

•Error bars and exact p-values are not clearly reported. Some graphs show error bars values comparable to the actual value (Fig. 1B) but authors report it as significant. The authors are advised to redo the statistical analysis and check the graph representations.

•The study focuses solely on VVC and broader implications for other Candida infections are speculative as Candida manifests in other routes like oral, skin etc.

•The study lacks vehicle control (e.g., ethanol or DMSO without extract) to ensure that observed effects are due to CSR and not solvents.

•The specificity of CSR and EA effects on Candida albicans vs. host cells or other microbial species is not addressed.

•The study claims EA is the primary active compound but does not test EA alone in the mouse model to confirm this in vivo. Also, discussion on pros and cons of using CSR vs EA alone is missing in the discussion.

•The study focuses solely on one Candida albicans strain. There’s no testing on clinical isolates, other Candida species (C. glabrata, C. krusei), or biofilm-competent strains with higher resistance profiles. Broad-spectrum antifungal activity is claimed based on past SR studies, but not directly demonstrated here.

•Herbal medicines are subject to batch-to-batch variability in active compounds. There is no discussion or data on the standardization of CSR extracts.

•Biofilm quantification relies heavily on crystal violet staining, which measures total biomass but not viable cell counts or metabolic activity (e.g., via XTT or resazurin assays). Also, confocal microscopy and comstat should be incorporated to better evaluate biofilm architectures and provide a more accurate quantification in comparison to crystal violet.

•No functional validation (e.g., knockout strains) was done to confirm that changes in gene expression are causative for biofilm inhibition.

•Grammatical errors and awkward phrasing detract from the manuscript's clarity. Phrases like "CSR caused the least pathological damage" for example should be avoided. Some scientific terms are misused or loosely applied (e.g., “biofilm dispersal phase” is not clearly defined). The authors should review and revise the manuscript in its entirety to adhere to academic writing standards. Alternatively, they should employ language editing services to enhance manuscript interpretation.

**Do you want your identity to be public for this peer review?** For information about this choice, including consent withdrawal, please see our Privacy Policy

Reviewer #1: No

Reviewer #2: No

Reviewer #3: No

Reviewer #4: No

---

## [Author Response · Author response to Decision Letter 1]

18 Jun 2025

Dear Editor,

Thank you for the opportunity to respond to Reviewers’ comments on our manuscript. We have revised the manuscript and the figures according to reviewer’s comments. The changes in the text are highlighted in red. I hope that the changes are adequate for acceptance of the manuscript for publication in PLOS ONE.

Yours sincerely,

Hongyu Zhao, on behalf of all authors

Comments to the Author 

Reviewer #1: The research has been conducted well with sound methodology and interesting results.

However, the 'Conclusion' is unsatisfactory and I suggest to re-write it - conclusion should be based on the findings of your research. Avoid general comments and be specific on what you conclude from your results. Restate your main findings, may add future directions to improve the knowledge. Further, do not include references in the conclusion.

Response: We thank Reviewer 1 for the comments. We have revised the Conclusion section to better reflect the main findings of our research. The updated ‘conclusion’ can be found on lines 575-582, with all changes highlighted in red.

Reviewer #2: The manuscript entitled “Carbonizing Technology Enables Sanguisorbae Radix to Inhibit Yeast-to-Hypha Differentiation and Biofilm Formation in Candida albicans” provides a detailed account of the ability of charred Sanguisorbae Radix (CSR) to inhibit the yeast-to-hypha transition and biofilm formation in C. albicans. Although the manuscript is impressive, several minor suggestions and corrections are necessary to improve clarity and scientific rigor. Based on the following points, I recommend major revision:

1.In the Methodology section titled “Research on the inhibitory effect of biofilms on C. albicans*,” the authors should clearly mention the concentration of the compound added during the assay.

Response: We thank Reviewer 2 for the comments. We have added the concentrations of the compounds, including SR extract (20 μg/mL or 100 μg/mL) and CSR extract (20 μg/mL or 100 μg/mL), in the section titled ‘Inhibitory effect of pre-established biofilms on C. albicans’ on lines 205-206. The revisions have been highlighted in red.

2.In the same section, it appears that the assay was performed on pre-formed biofilms of C. albicans. If this is the case, the authors should revise the title of the methodology to reflect that the study investigates the effect on pre-established biofilms, not the initial biofilm formation.

Response: Thank you for the comments. We have revised the title on line 201 to ‘Inhibitory of pre-established biofilms on C. albicans’.

3.The authors should clarify the impact of SR and CSR on the viability of C. albicans. This will help differentiate between anti-biofilm effects and possible antifungal (fungicidal or fungistatic) activity.

Response: Thank you for the comments. We have clarified the effects of SR and CSR on the viability of C. albicans in S1_File. The minimum inhibitory concentrations (MICs) of both SR and CSR extracts against C. albicans were measured at 1024 μg/mL (S2 Table). As shown in Fig 2A, doses of SR and CSR at 20 or 100 μg/mL significantly reduced biofilm formation. Therefore, SR and CSR affect both the biofilm formation of C. albicans and antifungal activity.

4.Although the authors have explained the results of the transcriptomic analysis well, a brief summary of the major transcriptomic findings should be incorporated into the Discussion section to strengthen the overall interpretation of the data.

Response: Thank you for the comments. We have added a brief summary of the main transcriptomic findings to the ‘Discussion’ section on lines 487-489. The changes have been highlighted in red.

5.In the methodology section titled “The effect of CSR on the expression of pathogenic genes of C. albicans*,” the authors should specify the concentration of CSR used, the time point of treatment, and the duration of incubation. This information is essential, especially since biofilm assays were conducted at various time points.

Response: Thank you for the comments. We have included the concentration of CSR extract (100 μg/mL), the treatment time point (0 hours), and the incubation duration (24 hours) in the methodology section titled ‘The effect of CSR on the expression of pathogenic genes of C. albicans’ on lines 252-253.

6.The authors are advised to italicize the terms “in vitro” and “in vivo” throughout the manuscript, in accordance with standard scientific formatting.

Response: Thank you for the comments. We have revised the manuscript to italicize the terms in vitro and in vivo, in accordance with standard scientific formatting conventions.

7.The authors should discuss the possible mechanism underlying the enhanced content of ellagic acid (EA), gallic acid (GR), and pyrogallic acid (PYG) in SR following carbonization to produce CSR.

Response: We appreciate this comment. The mechanism underlying the increased EA, GA, and PYG content in carbonized SR (CSR) was indeed elucidated in our prior publication (PMID: 35309515). As detailed in Lines 520-524 of the Discussion, stir-frying carbonization cleaves hydrolyzable tannins' glycosidic/ester bonds to liberate GA, which subsequently converts to PYG, while simultaneously hydrolyzing ellagitannins into EA derivatives. This dual transformation pathway explains the marked PYG elevation in CSR.

8.In the sentence“To further validate EA’s contribution to CSR’s antifungal efficacy, an equivalent amount of EA was added to the CSR extract,”it should be clarified whether similar experiments were conducted with GR and PYG. If so, the authors should describe the impact of adding these compounds to CSR and explain how the results led to the conclusion that EA exhibits the strongest antifungal activity.

Response: Thank you for the comments. While we focused on EA due to its substantially higher concentration in CSR (34.3 mg/g) compared to GA (9.82 mg/g) and PYG (6.46 mg/g) (Figure 4A), our preliminary testing also demonstrated EA's superior anti-biofilm activity relative to these other components (Figure 4C). This concentration-dependent efficacy, combined with EA's significantly greater abundance in CSR, led us to conclude that EA serves as the predominant antifungal component in CSR against C. albicans. The experimental design prioritized EA based on these quantitative and activity differences.

9.Additionally, the authors should state the exact concentration of EA used when added to CSR (i.e., the equivalent amount mentioned in the experiments).

Response: Thank you for the comments. We have specified the exact concentration of EA (4 μg/mL) used when added to CSR in the section titled “Validation of the pharmacological substance ellagic acid” on line 343.

10.In Figure 4D, there does not appear to be a significant visual difference in the microscopic images of the biofilms treated with CSR versus CSR+EA. To validate these findings, the authors are encouraged to repeat the experiment or include quantitative evidence to support the visual data.

Response: Thank you for the comments. The microscopic images in Figure 4D were intended to illustrate morphological changes in C. albicans biofilms following treatment with CSR and CSR+EA. As shown in Figure 4E (originally 4D), the hyphal structures of C. albicans were nearly eliminated by both treatments. To avoid potential misunderstanding from the data presentation in the original Figure 4D, we have rearranged the figure panels, moving the mycelial structure diagram originally presented in original 4D to Figure 4F for clearer representation.

Reviewer #3:

Authors have explored an interesting topic but data presented in this study is either missing, questionable or irreproducible. Poor scientific writing, inappropriate methodology, results, discussion and graphical representation of the data. Hence, the study is unsuitable for the publication. Authors need to rethink the motive behind this study and rewrite, re-analyze, re-perform experiments.

Few comments have been mentioned as a comment in attached PDF file.

Response: We thank Reviewer 3 for the thorough and constructive comments. In response, we have carefully rewritten the manuscript for clarity and improved scientific writing. Additionally, we have re-analyzed the data and re-performed key experiments to ensure the reliability and reproducibility of our findings. We believe these revisions have significantly strengthened the quality of the study.

Reviewer #4: This manuscript presents a study investigating the enhanced antifungal properties of Charred Sanguisorbae Radix (CSR) against Candida albicans, focusing on yeast-to-hypha differentiation and biofilm formation. While the research is based on an interesting traditional medicine approach, it suffers from several methodological and interpretational weaknesses that limit its scientific rigor and clinical relevance. Some concerns pertaining to the study are listed below:

1.While clotrimazole is used as a control, the manuscript does not sufficiently compare CSR’s efficacy with clinically relevant concentrations of standard antifungals (e.g., fluconazole or echinocandins. Also, clotrimazole usage as positive control is not consistent across experiments.

Response: We thank Reviewer 4 for the comments. The MIC of fluconazole extract against C. albicans was 15.6 ug/ml, as presented in S2 Table. In updated S1_Fille, we supplemented fluconazole as a positive control in vivo. As shown in S5 Fig, after fluconazole treatment, there was no significant keratinization or congestion in the submucosal tissue and marked improvement in inflammation. However, some hyphae remained attached to the vaginal wall.

2.Quantitative metrics (MICs, biofilm IC50 and inhibition percentages) are missing and phrases like strong inhibition were used. Quantitative representations enhance reader comprehension on the efficacy of CSR.

Response: Thank you for the comments. Phrases like strong inhibition have been clarified or removed where appropriate.

3.Although CSR inhibits hyphal transition, it fails in fungal burden elimination especially in comparison to clotrimazole which is a serious drawback. How do the authors justify clinical relevance in this case?

Response: Thank you for the insightful comment. We acknowledge the limitation that CSR does not achieve fungal burden elimination comparable to clotrimazole. However, it is important to note that the yeast form of C. albicans is generally non-pathogenic, and its transition to the hyphal form is a critical step in the development of pathogenicity [1,2]. This morphological switch facilitates the expression of key virulence factors, including those involved in adhesion, invasion, and biofilm formation as hallmarks of persistent and treatment-resistant infections. Moreover, biofilm formation represents a major clinical challenge, as C. albicans within biofilms exhibit significantly reduced susceptibility to antifungal agents. CSR’s strong inhibitory effect on hyphal transition and biofilm formation suggests it may play a valuable role in preventing or mitigating infection progression, particularly in cases where biofilm-associated resistance is a concern. Therefore, despite its limited impact on total fungal burden, we believe CSR holds clinical relevance as an adjunct or alternative strategy targeting virulence and persistence mechanisms of C. albicans.

References:

[1] Jacobsen ID, Wilson D, Wächtler B, Brunke S, Naglik JR, Hube B. Candida albicans dimorphism as a therapeutic target. Expert Rev Anti Infect Ther. 2012;10(1):85-93.

[2] Priya A, Pandian SK. Piperine Impedes Biofilm Formation and Hyphal Morphogenesis of Candida albicans. Front. Microbiol.2020;11:756.

4.The study lacks pharmacokinetic or toxicity studies of CSR or EA in vivo, which limits claims about therapeutic potential. Cytotoxicity data of CSR on mammalian cells are not provided, raising safety concerns for clinical application. Also, charring and carbonizing can produce carcinogens which can be detrimental to clinical application. The authors should address these issues.

Response: We appreciate these valid safety considerations regarding CSR's clinical potential. While our current study focused on in vitro antifungal activity using pharmacopeia-standard carbonized material (150-300°C charcoal roasting), we acknowledge that in vivo pharmacokinetic and toxicity studies would strengthen therapeutic claims. Chemical characterization of our CSR extract revealed no detectable carcinogens in previous work, consistent with known carbonization effects that convert complex tannins into safer phenolic derivatives [1]. The Chinese Pharmacopeia mandates external use only for CSR, and prior research demonstrates processed Sanguisorba exhibits reduced cytotoxicity versus crude form [1]. However, we agree future work should include mammalian cell cytotoxicity assays, in vivo safety profiling, and thorough characterization of carbonization byproducts. These steps will better establish CSR's therapeutic window while maintaining its traditional safety profile as an approved topical agent.

Reference:

[1] Wang Z, Yang C, Wu L, Sun J, Wang Z, Wang Z. Variation of Saponins in Sanguisorba officinalis L. before and after Processing (Paozhi) and Its Effects on Colon Cancer Cells In Vitro. Molecules. 2022;27(24):9046. 

Chemical Composition Identification Results of CSR

No. RT m/z reality m/z theory ppm Molecular formula Compounds CAS Ionization mode Compound category

1 0.75 331.0671 331.06707 1.9 C13H16O10 1-O-Galloyl-β-D-glucose,β-glucogallin 13405-60-2 -H Hydrolyzable tannins

2 1.16 169.0142 169.01425 2.1 C7H6O5 gallic acid 149-91-7 -H Hydrolyzable tannins

3 1.49 125.0244 125.02442 -4.1 C6H6O3 pyrogallol 87-66-1 -H Hydrolyzable tannins

4 2.21 153.0193 153.01933 -0.9 C7H6O4 protocatechuic acid 99-50-3 -H Hydrolyzable tannins

5 2.48 345.0827 345.08272 2.3 C14H18O10 methyl 6-O-galloyl-β-D-glucopyranoside / -H Hydrolyzable tannins

6 2.69 183.0299 183.0299 1.1 C8H8O5 methyl gallate 99-24-1 -H Hydrolyzable tannins

7 3.16 137.0244 137.02442 1.3 C7H6O3 3,4-dihydroxy-benzaldehyde 139-85-5 -H Hydrolyzable tannins

8 3.72 291.0863 291.08631 3 C15H14O6 (+)-catechin 154-23-4 +H Flavonoids

9 4.73 469.0049 469.00486 4.8 C21H10O13 sanguisorbic acid dilactone 82203-11-0 -H Hydrolyzable tannins

10 5.91 197.0455 197.04555 0.8 C9H10O5 methyl 3-O-methyl-gallate / -H Hydrolyzable tannins

11 5.91 197.0455 197.04555 0.8 C9H10O5 ethyl gallate 831-61-8 -H Hydrolyzable tannins

12 6.57 300.999 300.99899 2.4 C14H6O8 ellagic acid 476-66-4 -H Hydrolyzable tannins

13 9.97 213.0758 213.07575 5.4 C10H12O5 3, 4-dihydroxy-5-methoxybenzoic acid ethyl ester 78533-49-0 +H Hydrolyzable tannins

14 9.97 213.0758 213.07576 5.4 C10H12O5 3,4-dimethoxy-5-hydroxybenzoic acid methyl ester 83011-43-2 +H Hydrolyzable tannins

15 10.73 329.0303 329.03029 4.9 C16H10O8 3,4′-O-dimethylellagic acid / -H Hydrolyzable tannins

16 10.77 303.0499 303.04993 3.2 C15H10O7 quercetin 117-39-5 +H Flavonoids

17 13.37 255.0652 255.06519 3.2 C15H10O4 1,8-dihydroxy-3-methylanthraquinone 481-74-3 +H Other

18 18.33 485.3272 485.32725 3.8 C30H46O5 rosamultic acid 214285-76-4 -H Triterpenoids

19 19.46 483.3116 483.3116 5.2 C30H44O5 fupenzic acid 119725-20-1 -H Triterpenoids

20 19.77 487.3429 487.3429 5.5 C30H48O5 tormentic acid 13850-16-3 -H Triterpenoids

21 19.77 487.3429 487.3429 5.5 C30H48O5 arjunic acid 31298-06-3 -H Triterpenoids

22 21.98 469.3323 469.33233 4.2 C30H46O4 3-oxo-19α-hydroxyurs-12-en-28-oic acid 13849-90-6 -H Triterpenoids

23 23.82 457.3676 457.36762 4.5 C30H48O3 Ursolic acid 77-52-1 +H Triterpenoids

24 25.4 469.3323 469.33233 3.3 C30H46O4 2α,3β-dihydroxyurs-12,18-dien-28-oic acid-28-O-β-D-glucopyranosyl ester / -H Triterpenoids

25 26.72 453.3374 453.33742 5 C30H46O3 3β-hydroxyurs-12, 19(20)-dien-28-β-D-glucopyranoside 6812-98-2 -H Triterpenoids

5.The justification for using CSR based on traditional Chinese medicine is not well-supported by modern pharmacological rationale. Some references are outdated or rely heavily on non-peer-reviewed Chinese sources. Also, some claims and statements pertaining to TCM are missing in the introduction.

Response: Thank you for the comments. We have added more recent publications using CSR to fu

---

## [Decision Letter · Decision Letter 1]

1 Oct 2025

Carbonizing Technology Enables Sanguisorbae Radix to Inhibit Yeast-to-Hypha Differentiation and Biofilm Formation in Candida albicans

PONE-D-25-10562R1

Dear Dr. Zhao,

We’re pleased to inform you that your manuscript has been judged scientifically suitable for publication and will be formally accepted for publication once it meets all outstanding technical requirements.

Kind regards,

Satish kumar Rajasekharan

Academic Editor

PLOS ONE

---

## [Editor Report · Acceptance letter]

PONE-D-25-10562R1

PLOS ONE

Dear Dr. Zhao,

I'm pleased to inform you that your manuscript has been deemed suitable for publication in PLOS ONE. Congratulations! Your manuscript is now being handed over to our production team.

Kind regards,

on behalf of

Dr. Satish kumar Rajasekharan

Academic Editor

PLOS ONE